

# Prevalence and metabolic risk factors of chronic kidney disease among a Mexican adult population: a cross-sectional study in primary healthcare medical units

Alfonso R. Alvarez Paredes[1], Anel Gómez García[2],
Martha Angélica Alvarez Paredes[3], Nely Velázquez[4],
Diana Cindy Ojeda Bolaños[5], Miriam Sarai Padilla Sandoval[6],
Juan M. Gallardo[7], Gerardo Muñoz Cortés[8], Seydhel Cristina Reyes Granados[9],
Mario Felipe Rodríguez Morán[9,10], Joaquin Tripp[10], Arturo Lopez Pineda[10,11] and
Cleto Alvarez Aguilar[1]

[1] Facultad de Ciencias Médicas y Biológicas ''Dr. Ignacio Chávez'', Universidad Michoacana de San Nicolás de Hidalgo, Morelia, Michoacán, Mexico

[2] Centro de Investigación Biomédica de Michoacán, Instituto Mexicano del Seguro Social, Morelia, Michoacán, Mexico

[3] Unidad Médica de Atención Ambulatoria/Unidad de Medicina Familiar Núm. 75, Instituto Mexicano del Seguro Social, Morelia, Michoacán, Mexico

[4] Unidad de Medicina Familiar Núm. 80, Instituto Mexicano del Seguo Social, Morelia, Michoacán, Mexico

[5] Unidad de Medicina Familiar Núm. 84, Instituto Mexicano del Seguro Social, Morelia, Michoacán, Mexico

[6] Unidad de Medicina Familiar Núm. 82, Instituto Mexicano del Seguro Social, Zamora, Michoacán, Mexico

[7] Unidad de Investigación Médica en Enfermedades Nefrológicas, Hospital de Especialidades, Centro Médico Nacional Siglo XXI, Instituto Mexicano del Seguro Social, Mexico City, Mexico

[8] Coordinación Auxiliar Médica de Investigación en Salud, Órgano de Operación Administrativa Desconcentrada, Instituto Mexicano del Seguro Social, Morelia, Michoacán, Mexico

[9] Centro de investigación y Asistencia en Tecnología y Diseño del Estado de Jalisco, A.C., Consejo Nacional de Humanidades, Ciencia y Tecnología, Guadalajara, Jalisco, Mexico

[10] Amphora Health, Morelia, Michoacán, Mexico

[11] Escuela Nacional de Estudios Superiores, Unidad Morelia, Universidad Nacional Autónoma de México, Morelia, Michoacán, Mexico

Corresponding author
Cleto Alvarez Aguilar,
cleto.alvarez@umich.mx

## ABSTRACT

**Introduction**. The intricate relationship between obesity and chronic kidney disease (CKD) progression underscores a significant public health challenge. Obesity is strongly linked to the onset of several health conditions, including arterial hypertension (AHTN), metabolic syndrome, diabetes, dyslipidemia, and hyperuricemia. Understanding the connection between CKD and obesity is crucial for addressing their complex interplay in public health strategies.

**Objective**. This research aimed to determine the prevalence of CKD in a population with high obesity rates and evaluate the associated metabolic risk factors.

**Material and Methods**. In this cross-sectional study conducted from January 2017 to December 2019 we included 3,901 participants of both sexes aged ≥20 years who were selected from primary healthcare medical units of the Mexican Social Security Institute (IMSS) in Michoacan, Mexico. We measured the participants' weight, height, systolic and diastolic blood pressure, glucose, creatinine, total cholesterol, triglycerides, HDL-c,

LDL-c, and uric acid. We estimated the glomerular filtration rate using the Collaborative Chronic Kidney Disease Epidemiology (CKD-EPI) equation.

**Results**. Among the population studied, 50.6% were women and 49.4% were men, with a mean age of 49 years (range: 23–90). The prevalence of CKD was 21.9%. Factors significantly associated with an increased risk of CKD included age $\geq$60 years (OR = 11.70, 95% CI [9.83–15.93]), overweight (OR = 4.19, 95% CI [2.88–6.11]), obesity (OR = 13.31, 95% CI [11.12–15.93]), abdominal obesity (OR = 9.25, 95% CI [7.13–11.99]), AHTN (OR = 20.63, 95% CI [17.02–25.02]), impaired fasting glucose (IFG) (OR = 2.73, 95% CI [2.31–3.23]), type 2 diabetes (T2D) (OR = 14.30, 95% CI [11.14–18.37]), total cholesterol (TC) $\geq$200 mg/dL (OR = 6.04, 95% CI [5.11–7.14]), triglycerides (TG) $\geq$150 mg/dL (OR = 5.63, 95% CI 4.76-6.66), HDL-c <40 mg/dL (OR = 4.458, 95% CI [3.74–5.31]), LDL-c $\geq$130 mg/dL (OR = 6.06, 95% CI [5.12–7.18]), and serum uric acid levels $\geq$6 mg/dL in women and $\geq$7 mg/dL in men (OR = 8.18, 95% CI [6.92–9.68]), ($p < 0.0001$). These factors independently contribute to the development of CKD.

**Conclusions**. This study underscores the intricate relationship between obesity and CKD, revealing a high prevalence of CKD. Obesity, including overweight, abdominal obesity, AHTN, IFG, T2D, dyslipidemia, and hyperuricemia emerged as significant metabolic risk factors for CKD. Early identification of these risk factors is crucial for effective intervention strategies. Public health policies should integrate both pharmacological and non-pharmacological approaches to address obesity-related conditions and prevent kidney damage directly.

**Subjects** Diabetes and Endocrinology, Epidemiology

**Keywords** Chronic kidney disease, Glomerular filtration rate, Overweight, Obesity, Dyslipidemia, Hyperuricemia, Risk factors, Prevalence

## INTRODUCTION

The clinical practice guidelines of the Kidney Disease Outcomes Quality Initiative (K/DOQI) define chronic kidney disease (CKD) as the presence of structural or functional alterations of the kidney with or without a decrease in the glomerular filtration rate (GFR), persisting for at least 3 months, independent of cause. CKD imposes high costs on public health systems and leads to poor outcomes (*Levey et al., 2002*; Chapter 1: Definition and classification of CKD 2013). However, the medical community has raised concerns about its definition, classification, and methods used to estimate GFR. These concerns prompted the publication of new international guidelines from the Kidney Disease Consortium: Improving Global Outcomes (KDIGO) in 2005 (*Levey et al., 2005*) and 2012 (*Becker et al., 2012*). These guidelines refine the K/DOQI guidelines, providing recommendations for diagnosing, evaluating, and treating CKD, while managing arterial hypertension (AHTN), and facilitating risk stratification.

Obesity is a systemic, chronic, progressive, and multifactorial disease characterized by abnormal and excessive fat accumulation, posing a significant global health risk with increasing prevalence worldwide (*World Health Organization, 2000*; *World Health Statistics, 2021*). Both CKD and obesity impose substantial socioeconomic costs on healthcare

systems, reduce life expectancy, and are risk factors for chronic diseases (*Luyckx, Tonelli & Stanifer, 2018*; *Swinburn et al., 2011*). It is projected that by 2030, the global population of obese individuals will reach 1.35 billion (*Kelly et al., 2008*). Moreover, current trends suggest that by 2030, 86.3% of adults in the United States will be overweight or obese (*Wang et al., 2008*), while in Mexico, the Organization for Economic Cooperation and Development predicts that 40% of the adult population will be obese (*Obesity Update, 2017*). The Mexican Ministry of Health estimates that the total cost of obesity was 240 billion pesos in 2017, a figure expected to rise to 272 billion by 2023 (*Secretaría de Salud, 2015*). Obesity is recognized as a contributing factor in the pathogenesis of various diseases, including arterial hypertension (AHTN), type 2 diabetes (T2D), metabolic syndrome (MS), dyslipidemia, and hyperuricemia (*Eknoyan, 2006*; *Flegal et al., 2002*; *Haque et al., 2019*; *Mahdavi-Roshan et al., 2020*).

On the other hand, arterial hypertension and metabolic syndrome, type 2 diabetes, dyslipidemia, and hyperuricemia are risk factors for the development and progression of CKD (*Kazancioğlu, 2013*; *Tsao et al., 2023*). Recent studies have linked obesity to the development and progression of CKD due to hemodynamic effects such as moderate to severe albuminuria, structural changes including increased kidney weight, mesangial expansion, and damage to podocytes, as well as pathological changes characterized by glomerulomegaly and glomerulosclerosis (*Rutkowski et al., 2006*; *Toto et al., 2010*; *Ritz, Koleganova & Piecha, 2011*; *Matthew, Okada & Sharma, 2011*; *Goumenos et al., 2009*; *Hunley, Ma & Kon, 2010*). Additionally, other studies have shown that obesity is an independent risk factor for the onset, progression, and poor response to treatment of CKD associated with AHTN, MS, and T2D, even after adjusting for a cluster of comorbidities as confounding variables (*Eknoyan, 2011*).

This research aimed to determine the prevalence of CKD in a population with high obesity rates, such as the Mexican population, and to evaluate the associated metabolic risk factors, particularly in the interplay with obesity.

## MATERIALS & METHODS

### Study design and participants

In this cross-sectional population study conducted from January 2017 to December 2019, participants aged ≥20 years of both sexes were selected from primary healthcare medical units of the Mexican Social Security Institute (IMSS) in the state of Michoacan, Mexico. Participants were invited either by telephone or through written invitations, where we explained the study's objectives. Upon consenting to participate, subjects were scheduled to visit the designated medical units at 7 a.m. They were instructed to wear comfortable clothing, undergo a fasting period of at least 8 h, and provide a first-morning urine sample for analysis.

The sample size for the finite population was calculated based on the number of patients with type 2 diabetes mellitus (T2D) registered in primary healthcare medical units of IMSS in Michoacan, Mexico ($N = 27,297$), with a confidence level of 95%, a prevalence of 31.81% for nephropathy, and a precision level of 1.5% (*Tripp et al., 2023*), resulting in a

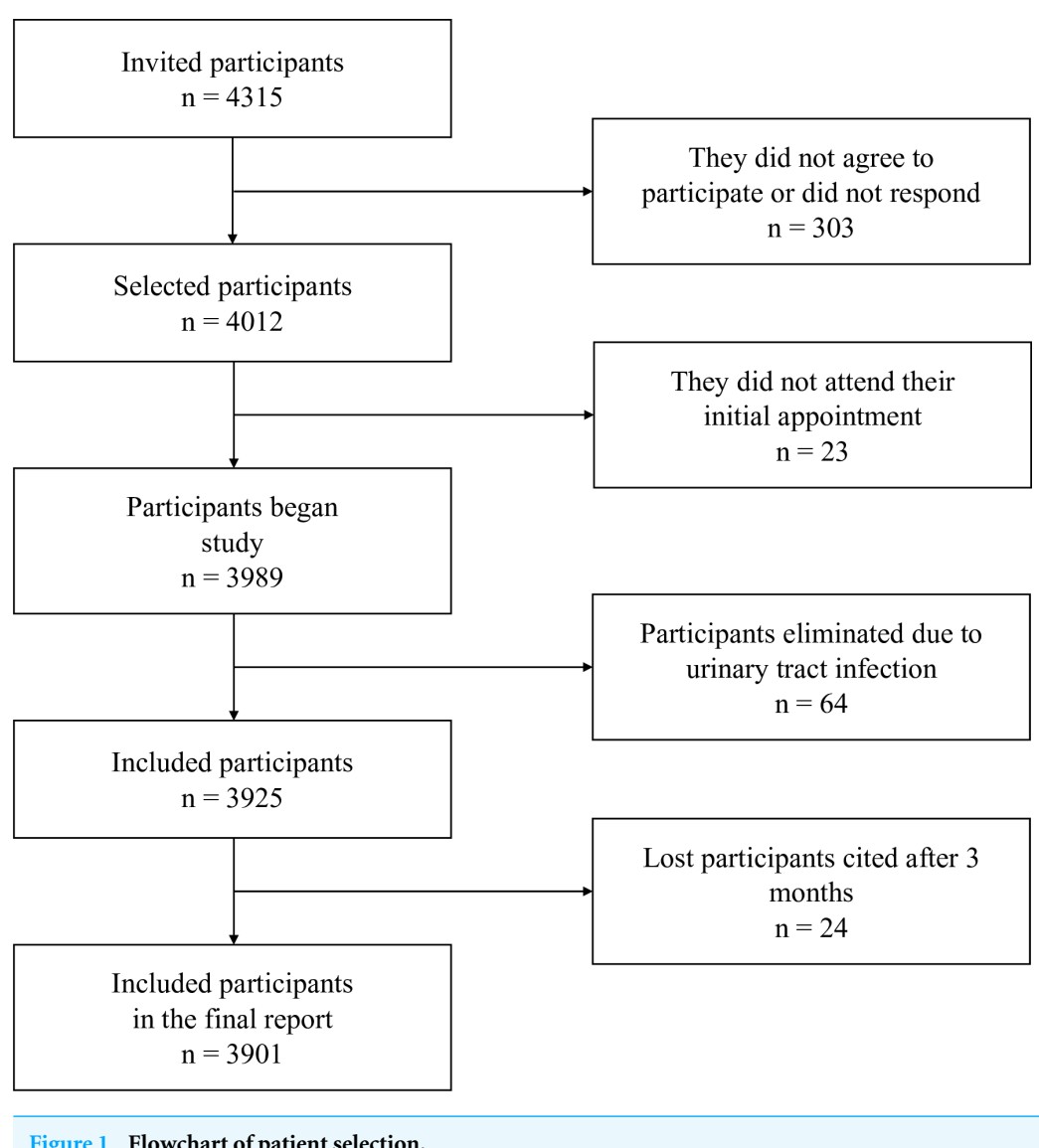

**Figure 1  Flowchart of patient selection.**

sample size of at least 3,262 patients. We included 3,923 patients. Figure 1 presents a flow chart of the study.

Out of the initially selected 4,012 participants, 23 did not attend their appointment, 64 were diagnosed with urinary tract infections and referred to medical care in their respective units, and 24 participants who did not attend their appointment within three months to confirm or rule out the presence of moderately or severely increased albuminuria were excluded. Therefore, results from 3,901 participants are presented. All participants provided both verbal and written consent before their inclusion in the study. Ethical approval for the research was granted by the IMSS National Ethics Commission, with Institutional Review Board (IRB) registration number R-2015-785-010.

## CLINICAL CHARACTERISTICS AND BIOCHEMICAL PROCEDURES

Anthropometric measurements, including weight and height, were taken while participants wore light clothing and no shoes, using a scale with a stadiometer that had been previously calibrated. Body Mass Index (BMI) was calculated using the Quetelet index formula (weight (kg)/height (m$^2$)). We adhered to the guidelines outlined in the American Association of Clinical Endocrinologists (AACE) and American College of Endocrinology (ACE) Comprehensive Clinical Practice Guidelines for Medical Care of Patients with Obesity (*Garvey et al., 2016*), which classify a BMI between 25 and 29.9 kg/m$^2$ as indicative of overweight and a BMI of $\geq$30 kg/m$^2$ as obesity. Abdominal obesity in the study population was determined by measuring waist circumference, with cutoff points established by the International Diabetes Federation (IDF) of $\geq$80 cm in women and $\geq$90 cm in men (*Alberti et al., 2009*).

Blood pressure was measured using previously calibrated OMRON HEM-7130 automatic blood pressure monitors (Omron Healthcare, Inc., Lake Forest, IL, USA). Measurements were taken in the antecubital region of the left arm, with participants in a seated position following a 5-minute rest period. Three blood pressure readings were obtained, with one-minute intervals between each recording. The initial measurement was discarded, and the average of the second and third readings was calculated for analysis. Arterial hypertension was defined as systolic blood pressure (SBP) $\geq$140 mmHg and diastolic blood pressure (DBP) $\geq$90 mmHg, in accordance with the Consensus on Systemic Arterial Hypertension in Mexico. Lower values were considered if participants were taking antihypertensive medications (*Rosas-Peralta et al., 2016*).

Before their interviews at each primary healthcare facility, participants were instructed to arrive in the morning after fasting for at least 8 h. They provided a sample of first-morning urine in a clean, dry bottle without preservatives, and a 7 ml blood sample was drawn from the right antecubital vein into a vacutainer tube without anticoagulant. Urine samples were used to measure the concentrations of albumin and creatinine at the time of collection. The blood samples were then centrifuged at 4,000 revolutions per minute for 10 min, and the resulting serum was stored in 500 μl aliquots in a −30 °C freezer for subsequent biochemical analysis.

The concentrations of albumin and creatinine in urine were determined using urinalysis strips capable of qualitatively and semi-quantitatively detecting low concentrations of creatinine and albumin (Ref. U031-021; ACON Laboratories, Inc., San Diego, CA, USA), and the albumin-to-creatinine ratio (ACR) was calculated. ACR values <30 mg/g were considered normal to mildly increased albuminuria, 30-300 mg/g indicated moderately increased albuminuria, and >300 mg/g indicated severely increased albuminuria. Urinary tract infections were also ruled out using urinalysis strips (Urine Reagent Strips; Untimed Products (Deutschland) GmbH, Ahrensburg, Germany), which detect nitrites and leukocytes indicative of urinary tract infections. Participants diagnosed with urinary infections *via* urinalysis strip were excluded from the sample and referred to their primary healthcare facility for appropriate care.

The GFR was estimated using the Collaborative Chronic Kidney Disease Epidemiology (CKD-EPI) equation (*Levey et al., 2009*; *Becker et al., 2012*). The stages of CKD were defined and classified according to the KDIGO guidelines, which incorporate the urinary albumin-to-creatinine ratio (ACR) and estimated glomerular filtration rate (eGFR). These stages include G1, G2, G3a, G3b, G4, and G5, along with three levels of albuminuria based on the ACR described earlier. Participants were classified as CKD patients if their eGFR was <60 ml/min/1.73 m$^2$ or if their eGFR was ≥60 ml/min/1.73 m$^2$ and their ACR was ≥30 mg/g persistently measured at two separate occasions three months apart.

Glucose, urea, creatinine, total cholesterol (TC), triglycerides (TG), low-density lipoprotein cholesterol (LDL-c), high-density lipoprotein cholesterol (HDL-c), and uric acid levels were determined using enzymatic colorimetric tests conducted on automated clinical chemistry equipment (Kontrol Lab Mod. ES-218). Impaired fasting glucose (IFG) was defined as fasting serum glucose (FSG) levels ranging from 100 to 125 mg/dL. Type 2 diabetes (T2D) was defined as FSG levels ≥126 mg/dL, following diagnostic criteria established by the American Diabetes Association in 2018 (*American Diabetes Association, 2018*).

Dyslipidemia was considered present if participants met one or more of the following criteria: TC ≥200 mg/dL, TG ≥150 mg/dL, LDL-c ≥130 mg/dL, or HDL-c <40 mg/dL, as per the NCEP-ATP-III guideline. Hyperuricemia was identified by serum uric acid concentrations ≥6 mg/dL in women and ≥7 mg/dL in men, as reported in the Diabetes Control and Complications Trial/Epidemiology of Diabetes Interventions and Complications Study (DCCT/EDIC) (*Jenkins et al., 2021*).

## Statistical analysis

Statistical analysis was performed using SPSS software version 22 for Windows, with a significance level set at $p < 0.05$. We summarized continuous variables using median, minimum, and maximum values while expressing categorical variables as percentages. The Kolmogorov–Smirnov test was employed to assess normal distribution. We evaluated differences in continuous variables using the Mann–Whitney U test and analyzed categorical variables with the chi-square test.

We conducted multivariate logistic regression to explore interactions among factors associated with CKD, defined by KDIGO 2012 risk categories (moderate to very high). CKD was the dependent variable, with independent variables including age ≥60 years, overweight, obesity, abdominal obesity, arterial hypertension (AHTN), impaired fasting glucose (IFG), type 2 diabetes (T2D), dyslipidemia, and hyperuricemia, adjusted for gender. The goodness-of-fit of this model was evaluated using the Hosmer-Lemeshow method. Collinearity was addressed using two methods: first, regression lineal tests considered a variance inflation factor (VIF) with a cutoff of 5 in the analysis, and second, a correlation matrix analysis discarded variables with absolute values higher than 0.8.

## RESULTS

The study population included 1,975 (50.6%) women and 1,926 (49.4%) men, among whom CKD was prevalent in 856 (21.9%) participants. Table 1 presents the clinical

**Table 1** Clinical and biochemical characteristics of participants according to CKD status.

| Characteristics | With CKD<br>n = 856 | Without CKD<br>n = 3045 | P value |
|---|---|---|---|
| **Clinical characteristics** | | | |
| Gender | | | |
|     Female | 448 (52.3) | 1,527 (50.1) | 0.137 |
|     Male | 408 (47.7) | 1,518 (49.9) | |
| Age (years) | 65 (43–90) | 44 (23–83) | <0.0001 |
| BMI (kg/m$^2$) | 31.4 (20.5–41.8) | 25.9 (17.0–39.3) | <0.0001 |
| Abdominal circumference (cm) | 99 (68–123) | 87 (63–119) | <0.0001 |
| Systolic BP (mm/Hg) | 140 (76–180) | 120 (85–190) | <0.0001 |
| Diastolic BP (mm/Hg) | 90 (60–110) | 80 (45–120) | <0.0001 |
| **Biochemical characteristics** | | | |
| Serum glucose (mg/dL) | 109.1 (55.8–295.6) | 87.0 (47.7–298.8) | <0.0001 |
| Serum creatinine (mg/dL) | 1.05 (0.72–9.80) | 0.80 (0.44–1.30) | <0.0001 |
| Total cholesterol (mg/dL) | 224.9 (73.5–393.2) | 188.3 (83.0–385.9) | <0.0001 |
| Triglycerides (mg/dL) | 188.3 (47.9–394.5) | 139.7 (32.1-397.00) | <0.0001 |
| HDL-cholesterol (mg/dL) | 44.0 (23.0–94.1) | 49.0 (22.0–98.2) | <0.0001 |
| LDL-cholesterol (mg/dL) | 129.7(53.1–199.2) | 101.1 (29.2–198.7) | <0.0001 |
| Uric acid (mg/dL) | 7.1 (3.2–10.5) | 5.2 (2.9–9.3) | <0.0001 |
| eGFR (ml/min per 1.73 m$^2$ | 68.9 (4.8–99.4) | 101.9 (60.1–140.7) | <0.0001 |
| Microalbuminuria (μg/mL) | 30.0 (10.0–1800.0) | 10.0 (10.0–29.0) | <0.0001 |

**Notes.**

CKD, Chronic Kidney Disease; BMI, Body Mass Index; BP, Blood Pressure; HDL, High Density Lipoprotein; LDL, Low Density Lipoprotein; eGFR, Estimated Glomerular Filtration Rate.

Gender is shown in percentages and continuous variables in median, minimum and maximum.

and biochemical characteristics of the study population based on renal function status. All clinical and biochemical characteristics showed statistically significant differences ($p < 0.0001$) across the groups, except for gender. Figure 2 illustrates the classification of CKD based on combined measures of albuminuria and eGFR to indicate prognosis. The low-risk category was observed in 3,045 (78.1%) participants, moderate risk in 569 (14.5%), high risk in 194 (5.0%), and very high risk in 93 (2.4%) participants. Notably, 19 (0.5%) participants exhibited severely decreased eGFR, and seven (0.2%) had a GFR <15 ml/min/1.73 m$^2$, indicative of renal failure. Among patients classified as moderately elevated risk and high risk (yellow and orange categories in Fig. 3), none had been previously diagnosed with kidney damage. Among patients classified as very high risk, only 17% had a prior diagnosis of CKD.

Table 2 demonstrates a significant association between age ≥60 years, overweight, obesity, abdominal circumference, AHTN, IFG, T2D, dyslipidemia, hyperuricemia, and CKD ($p < 0.0001$). Surprisingly, 504 (61.4%) patients with overweight and obesity had an eGFR ≥60 and ACR ≥30. Similarly, AHTN showed a direct association with varying degrees of kidney function decline; among hypertensive patients, 767 (65.6%) were previously diagnosed with hypertension, with 22 (81.1%) regularly attending clinics for antihypertensive treatment, while 145 (18.9%) did not receive treatment. Among

| Prognosis of CKD by GFR and Albuminuria Categories | | | | Albuminuria categories Description and range | | |
|---|---|---|---|---|---|---|
| | | | | A1 | A2 | A3 |
| | | | | Normal or slightly elevated | Moderately elevated | Severely elevated |
| | | | | <30 mg/g | 30–300 mg/g | >300 mg/g |
| eGFR categories (ml/min/1.73 m²) Description and range | G1 | Normal or high | >90 | * 2,200 56.4 % | £ 29 0.7 % | ¢ |
| | G2 | Slightly decreased | 60–90 | * 845 21.7 % | £ 495 12.7 % | ¢ 6 0.2 % |
| | G3a | Low-moderate decrease | 45–59 | £ 45 1.1 % | ¢ 188 4.8 % | § 19 0.5 % |
| | G3b | Moderate-severe decrease | 30–44 | ¢ | § 37 0.9 % | § 11 0.3 % |
| | G4 | Severe decrease | 15–29 | § | § 8 0.2 % | ∂ 11 0.3 % |
| | G5 | Renal failure | <15 | ∂ | ∂ | ∂ 7 0.2 % |

| | |
|---|---|
| * Low risk (if no other markers of kidney disease, no CKD)–green | ¢ Moderately high risk–orange |
| £ Moderate risk–yellow | § High risk–red |
| | ∂ Very high risk – deep red |

KDIGO 2012.

**Figure 2** Classification of chronic kidney disease to indicate prognosis based on the combined measures of albuminuria and estimated glomerular filtration rate. CKD, chronic kidney disease; GFR, glomerular filtration rate; KDIGO, Kidney Disease: Improving Global Outcomes.

hypertensive patients receiving treatment, 462 (60.2%) had well-controlled blood pressure levels. Regarding IFG and T2D, only 213 (57.7%) participants with T2D reported receiving treatment with hypoglycemic agents, while only 130 (15.4%) participants with IFG were undergoing non-pharmacological measures.

Figure 3 displays the outcomes of logistic regression analysis, identifying noteworthy independent risk factors associated with CKD development. These include age $\geq 60$ years (OR = 11.71, 95% CI [9.84–15.94], $p < 0.0001$), overweight (OR = 4.20, 95% CI [2.88–6.11], $p < 0.0001$), obesity (OR = 13.31, 95% CI [11.12–15.94], $p < 0.0001$), abdominal obesity (OR = 9.26, 95% CI [7.14–11.99], $p < 0.0001$), AHTN (OR = 20.64, 95% CI [17.02–25.02], $p < 0.0001$), IFG (OR = 2.74, 95% CI [2.31–3.24], $p < 0.0001$), T2D (OR = 14.31, 95% CI [11.14–18.37], $p < 0.0001$), hypercholesterolemia (OR = 6.04, 95% CI [5.11–7.14], $p < 0.0001$), hypertriglyceridemia (OR = 5.63, 95% CI [4.76–6.66], $p < 0.0001$), LDL-c $\geq 130$ mg/dL (OR = 6.07, 95% CI [5.13–7.18], $p < 0.0001$), HDL-c <40 mg/dL (OR = 4.46, 95% CI [3.74–5.31], $p < 0.0001$), and hyperuricemia (OR = 8.19, 95% CI [6.93–9.68], $p < 0.0001$). These factors independently predict CKD development

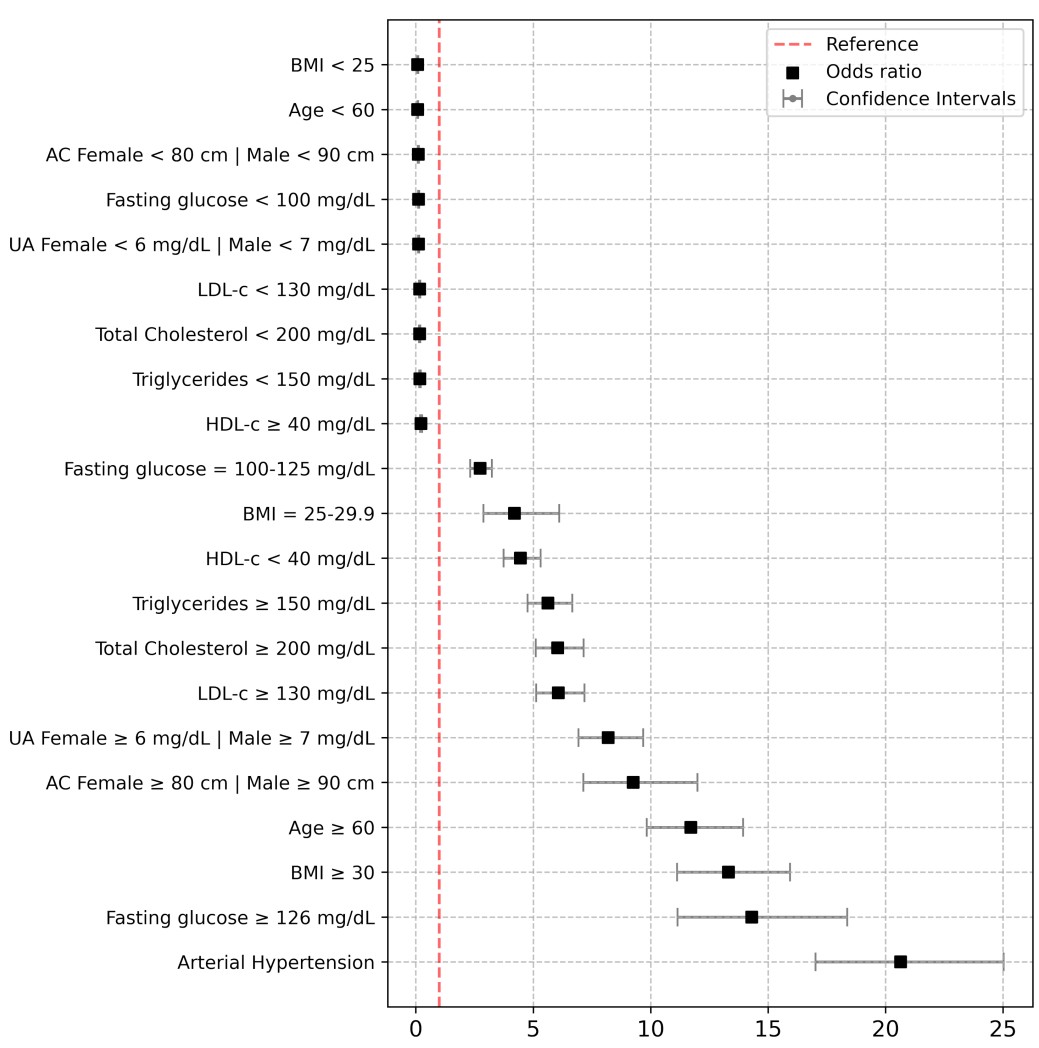

**Figure 3** Risk assessment of BMI, age, abdominal circumference, uric acid, total cholesterol, triglycerides, LDL-c, HDL-c, fasting glucose and arterial hipertension. Risk assessment of BMI (<25, 25–29.9, ≥30); age (<60, ≥60) years; Abdominal circumference (AC) (female <80, male <90, and female ≥80, male ≥90 centimeters); uric acid (UA) (female <6, male <7 and female ≥6, male ≥7 mg/dL); total cholesterol (<200 and ≥200 mg/dL); triglycerides (<150 and ≥150 mg/dL); low density lipoprotein cholesterol (LDL-c) (<130 and ≥130 mg/dL); high density lipoprotein cholesterol (HDL-c) (<40 and ≥40 mg/dL); fasting glucose (<100, 100–125 and ≥126 mg/dL), arterial hipertension (systolic blood pressure (SBP) ≥140 mmHg and diastolic blood pressure (DBP) ≥90 mmHg, or lower values if antihypertensive drugs were taken). Are variables were adjusted by sex.

significantly. The Hosmer-Lemeshow test showed good model calibration, with a chi-squared of 8.47, 7 degrees of freedom, and a significance of 0.293 (Supplementary Material 1–3).

## DISCUSSION

Chronic kidney disease (CKD) has emerged as a significant global public health concern due to its escalating prevalence, association with cardiovascular disease (CVD) mortality,

**Table 2  Association of CKD with overweight and obesity, abdominal circumference, arterial hypertension, impaired fasting glucose, type 2 diabetes, dyslipidemia and hyperuricemia.**

| | Total frequence n (%) | Participants with CKD n (%) | Participants without CKD n (%) | P value |
|---|---|---|---|---|
| Age (years) | | | | |
| ≥60 | 1,115 (28.6) | 674 (78.7) | 441 (14.5) | <0.0001 |
| <60 | 2,786 (71.4) | 182 (21.3) | 2,604 (85.5) | |
| BMI (kg/m$^2$) | | | | |
| <25 | 1,228 (31.5) | 35 (4.1) | 1,193 (39.2) | <0.0001 |
| 25–29.9 | 1,447 (37.1) | 157 (18.3) | 1,290 (42.4) | |
| ≥30 | 1,226 (31.4) | 664 (77.6) | 562 (18.4) | |
| Abdominal circumference | | | | |
| ≥80 Women, ≥90 Men | | | | |
| YES | 2,562 (65.7) | 801 (93.6) | 1,761 (57.8) | <0.0001 |
| NO | 1,339 (34.3) | 55 (6.4) | 1,284 (42.2) | |
| Arterial hypertension (mmHg) | | | | |
| YES | 1,168 (29.9) | 682 (79.7) | 486 (16.0) | <0.0001 |
| NO | 2,733 (70.1) | 174 (20.3) | 2,559 (84.0) | |
| Fasting serum glucose (mg/dL) | | | | |
| <100 | 2,703 (69.3) | 276 (32.2) | 2,427 (79.7) | <0.0001 |
| 100–125 | 833 (21.3) | 310 (36.3) | 523 (17.2) | |
| ≥126 | 365 (9.4) | 270 (31.5) | 95 (3.1) | |
| Dyslipidemia | | | | |
| TC ≥200 mg/dL | | | | |
| YES | 1,487 (38.1) | 608 (71.0) | 879 (28.9) | <0.0001 |
| NO | 2,414 (61.9) | 248 (29.0) | 2,166 (71.1) | |
| TG ≥150 mg/dL | | | | |
| YES | 1,579 (40.5) | 618 (72.2) | 961 (31.6) | <0.0001 |
| NO | 2,322 (59.5) | 238 (27.8) | 2,084 (68.4) | |
| LDL-c ≥130 mg/dL | | | | |
| YES | 703 (18.0) | 329 (38.4) | 374 (12.3) | <0.0001 |
| NO | 3,198 (82.0) | 527 (61.6) | 2,671 (87.7) | |
| HDL-c <40 mg/dL | | | | |
| YES | 848 (21.7) | 424 (49.5) | 424 (13.9) | <0.0001 |
| NO | 3,053 (78.3) | 432 (50.5) | 2,621 (86.1) | |
| Hyperuricemia | | | | |
| >6 mg/dL in women and >7 mg/dL in men | | | | |
| YES | 1,085 (27.8) | 546 (63.8) | 540 (17.7) | <0.0001 |
| NO | 2,816 (72.2) | 310 (36.2) | 2,505 (82.3) | |

**Notes.**

Values are shown in number and percentages. Normal renal function was considered to be one estimated Glomerular Filtration Rate ≥60 ml/min/1.73 m$^2$ and ACR<30 mg/g.

BMI, body mass index; CKD, Chronic kidney disease; TC, Total cholesterol; TG, triglycerides; HDL-c, High-density lipoprotein cholesterol; LDL-c, Low-density lipoprotein cholesterol.

adverse outcomes, severe complications, and substantial economic burdens, all contributing to diminished quality of life and strain on healthcare systems (*GBD Chronic Kidney Disease Collaboration, 2020*). Among the most prevalent risk factors are elderly age, overweight and obesity, abdominal obesity, arterial hypertension (AHTN), impaired fasting glucose (IFG), type 2 diabetes (T2D), dyslipidemia, and hyperuricemia (*Hunley, Ma & Kon, 2010*; *Eknoyan, 2011*; *Kovesdy, 2022*; *Kuma & Kato, 2022*). In this study conducted in an cohort of the Mexican adult population, we observed a CKD prevalence of 21.9%. Previous investigation conducted by the National Kidney Foundation Kidney Early Evaluation Program (KEEP) (a pilot phase of KEEP Mexico began in 2008), which is a free community screening program aimed at early detection of kidney diseases conducted in the Mexican population (Mexico City and the State of Jalisco) also reported a high prevalence of CKD (22% and 33% respectively) (*Obrador et al., 2010*).

Advanced age is recognized as a risk factor for declining kidney function due to the natural aging process affecting all human organs, including the kidneys. Consistent with findings from other studies in primary care settings (*Salvador-Gonzalez et al., 2017*), we observed a direct correlation between age ≥60 years and kidney function decline. This association becomes more pronounced in the presence of other concurrent CKD risk factors such as arterial hypertension (AHTN) and type 2 diabetes (T2D). This finding holds particular significance considering the ongoing intense and accelerated aging of the Mexican population, as reported by the National Institute of Statistics and Geography (INEGI) in press release number 568/22 dated September 30, 2022 (*Instituto Nacional de Estadística y Geografía, 2022*). This demographic trend suggests that the burden of kidney-related issues may worsen in the foreseeable future.

In our study, arterial hypertension (AHTN) was identified in 29.9% of participants, with 34.4% of them unaware of their hypertensive status. Among those aware of their hypertension, 81.1% received antihypertensive treatment, and 60.2% achieved controlled blood pressure levels. ENSANUT 2022 similarly reported a prevalence of AHTN at 30.1% (*Campos-Nonato et al., 2023*), which is higher than the 25.5% prevalence reported in ENSANUT 2016 (*Campos-Nonato et al., 2018*). Among undiagnosed hypertensive individuals, 40% were unaware of their condition, whereas among those diagnosed, 79.3% received antihypertensive treatment. These findings are not exclusive to the Mexican population, as demonstrated by a comprehensive study in primary care units across 200 countries and territories from 1990 to 2019, which highlighted similar trends in AHTN prevalence, detection, treatment, and control (*NCD Risk Factor Collaboration, 2022*). AHTN showed a direct association with various degrees of kidney function decline in our study, underscoring the importance of prioritizing both non-pharmacological and pharmacological strategies with nephroprotective effects.

Regarding type 2 diabetes (T2D), we observed a prevalence of 9.4%, whereas ENSANUT reported higher prevalences of 13.7% in 2016 (*Basto-Abreu et al., 2020*) and 12.6% in 2022 (*Campos-Nonato et al., 2023*). Additionally, 21.3% of participants had impaired fasting glucose (IFG), with CKD documented in 66.1% of T2D patients and 32.6% of those with IFG. These findings are significant given the rising prevalence of diabetes in Mexico. Without policy changes, the incidence of diagnosed diabetes is projected to increase

disproportionately, potentially affecting between 15 and 25 million adults by 2050, with a lifetime risk estimated at one in three to one in two (*Meza et al., 2015*).

In our study cohort, we found that overweight, obesity, and abdominal obesity are metabolic risk factors associated with declining kidney function. Notably, overweight and obese participants without other pathological conditions showed early signs of renal damage (eGFR $\geq$90 ml/min/m$^2$ and ACR $\geq$30 mg/g). Recent research has highlighted overweight and obesity as independent risk factors for CKD progression, irrespective of underlying conditions (*Tsuboi & Okabayashi, 2021*; *Jiang et al., 2023*; *Hojs et al., 2023*; *Bae et al., 2021*). These findings are particularly significant in the context of Mexico, where overweight and obesity are highly prevalent. According to the *Instituto Nacional de Estadística y Geografía (2020)*, Mexico ranks second globally in overweight and obesity, trailing only the United States, posing a major public health concern. Additionally, data from the National Health and Nutrition Survey (ENSANUT) (*Campos-Nonato et al., 2023*) show that 75.2% of Mexican adults are overweight or obese, with 81.0% classified as abdominally obese—a slight increase from ENSANUT 2016, indicating a troubling rise in obesity rates. Furthermore, a microsimulation study projects that by 2050, only 9% of Mexican adults will maintain a normal weight, while 34% will be overweight, and a staggering 57% will be obese. This trend is expected to have a significant impact on both the health and economic burden of the Mexican healthcare system (*Rtveladze et al., 2014*).

To our knowledge, this study is the first conducted in the Mexican adult population to establish a direct and independent association between overweight, obesity, and abdominal obesity with kidney function decline. Although the exact pathophysiological mechanisms remain unclear, they are likely linked to hemodynamic factors. Importantly, the presence of glomerular hyperfiltration syndrome is notable, potentially explaining the observed moderate increase in microalbuminuria due to heightened functional demands resulting from increased body mass and abdominal adiposity. This response may signify an adaptation to increased renal plasma flow without a corresponding adjustment in nephron count (*Chagnac et al., 2000*).

In this study, altered lipid and uric acid concentrations emerged as independent risk factors for CKD. According to our definition, 50.3% of the participants exhibited dyslipidemia, while 27.8% had hyperuricemia. Among them, a significant proportion—71.0% with hypercholesterolemia, 72.2% with hypertriglyceridemia, 38.4% with high LDL-c, 49.5% with low HDL-c, and 63.8% with elevated uric acid levels—were found to have some degree of kidney damage. These findings are significant given that cardiovascular disease (CVD) is the leading cause of mortality among individuals with end-stage kidney disease (*Bhandari et al., 2022*). One proposed pathophysiological mechanism involves the potential widespread and accelerated formation of atherosclerotic plaques due to hyperlipidemia, uremic toxins, inflammation, oxidative stress, and endothelial dysfunction (*Bulbul et al., 2018*). These comorbidities are often associated with overweight and obesity, further increasing the risk of CVD morbidity and mortality (*Serrano, Shenoy & Cantarin, 2023*).

The deleterious effects of obesity are closely associated with comorbidities such as T2D and AHTN. However, adipose tissue functions as an endocrine organ, exerting

a direct influence on the kidneys by secreting various hormones including leptin, adiponectin, resistin, pro-inflammatory cytokines (such as Interleukin-6 and Tumor Necrosis Factor-$\alpha$), and components of the Renin-Angiotensin-Aldosterone System (RAAS). These factors contribute to pathophysiological alterations such as activation of the sympathetic nervous system (SNS), systemic RAAS activation, vasoconstriction, hypertension, inflammation, disturbances in lipid metabolism, and oxidative stress (*Kershaw & Flier, 2004*). These hormones, cytokines, proteins, and systems collectively play critical roles in the development and progression of CKD.

Our logistic regression model, adjusted for sex, confirms that overweight, obesity, abdominal obesity, AHTN, T2D, dyslipidemia, and hyperuricemia are independent risk factors for CKD. This finding aligns with our previous research conducted among Mexican adults, which identified T2D and AHTN as independent contributors to kidney function decline, alongside other factors influenced by social inequality (*Amato et al., 2005*). Importantly, our study revealed that individuals with IFG already show signs of renal function decline. This suggests that even in the early stages of diabetes, changes in renal function may occur due to insulin resistance or hyperinsulinism, potentially exacerbating conditions like sympathetic nervous system hyperactivity, diabetes, and AHTN.

The emergence of lifestyle-related diseases such as insulin resistance, obesity, abdominal obesity, AHTN, IFG, T2D, dyslipidemia, and hyperuricemia are components of metabolic syndrome (MS), which in turn increases the risk of incident CKD (*Mirabelli et al., 2020*). The high prevalence of obesity observed across various age groups in the Mexican population underscores the urgency of addressing these risk factors to mitigate the development and progression of CKD from an early age (*Shamah-Levy et al., 2023*; *Campos-Nonato et al., 2023*). Therefore, promoting lifestyle changes should be a key focus to reduce the burden of CKD in the Mexican population (*Kuma & Kato, 2022*).

The study has several limitations. One limitation is that this type of study can establish associations but not causality. However, our robust results suggest that causality could potentially be established through studies employing different methodologies that analyze these same factors. Another limitation is the inclusion of only the adult population from a single state in Mexico and participants assigned to first-level medical units of a single health institution, which limits the generalizability to the entire Mexican population. Nonetheless, this limitation may also be considered a strength, as the IMSS serves more than half of Mexico's population. Previous studies conducted by us *Amato et al. (2005)* and by *Obrador et al. (2010)* in densely populated regions like Mexico City and the State of Jalisco have also reported a high prevalence.

## CONCLUSIONS

The Mexican adult population bears a significant burden of CKD, ranging from early stages to End-Stage Renal Disease. Metabolic risk factors such as overweight, obesity, abdominal obesity, AHTN, IFG, T2D, dyslipidemia, and hyperuricemia independently contribute to kidney function loss. Obesity plays a pivotal pathophysiological role in the development of these risk factors for both CKD and CVD. Early compromise of renal

structure and function is associated with increased adiposity during obesity. Hence, early identification of these metabolic risk factors and implementation of pharmacological and non-pharmacological interventions are crucial for mitigating the development and progression of kidney damage.

## ACKNOWLEDGEMENTS

We extend our gratitude to the volunteers who participated in this study. We trust that these findings will inform health policymakers, enabling informed decisions that benefit the population from a preventive standpoint.

### Funding

This study was supported with the financing granted in the extraordinary call 2017 to obtain financial support for the development of health research protocols from the Coordination of Health Research FIS/IMSS/PROT/G17-2/1719 of the Instituto Mexicano Seguro Social. The funders had no role in study design, data collection and analysis, decision to publish, or preparation of the manuscript.

### Grant Disclosures

The following grant information was disclosed by the authors:
Coordination of Health Research: FIS/IMSS/PROT/G17-2/1719.

### Competing Interests

Author ALP holds shares of Amphora Health. Authors ALP, JT, and MFRM contributed to the research while employed by Amphora Health. All other authors declare no competing interests.

### Author Contributions

- Alfonso R. Alvarez Paredes conceived and designed the experiments, performed the experiments, analyzed the data, prepared figures and/or tables, authored or reviewed drafts of the article, and approved the final draft.
- Anel Gómez García conceived and designed the experiments, analyzed the data, prepared figures and/or tables, authored or reviewed drafts of the article, and approved the final draft.
- Martha Angélica Alvarez Paredes performed the experiments, authored or reviewed drafts of the article, and approved the final draft.
- Nely Velázquez performed the experiments, authored or reviewed drafts of the article, and approved the final draft.
- Diana Cindy Ojeda Bolaños performed the experiments, authored or reviewed drafts of the article, and approved the final draft.
- Miriam Sarai Padilla Sandoval performed the experiments, authored or reviewed drafts of the article, and approved the final draft.

- Juan M. Gallardo analyzed the data, authored or reviewed drafts of the article, and approved the final draft.
- Gerardo Muñoz Cortés conceived and designed the experiments, performed the experiments, analyzed the data, authored or reviewed drafts of the article, and approved the final draft.
- Seydhel Cristina Reyes Granados conceived and designed the experiments, performed the experiments, authored or reviewed drafts of the article, and approved the final draft.
- Mario Felipe Rodríguez Morán conceived and designed the experiments, performed the experiments, authored or reviewed drafts of the article, and approved the final draft.
- Joaquin Tripp analyzed the data, prepared figures and/or tables, authored or reviewed drafts of the article, and approved the final draft.
- Arturo Lopez Pineda conceived and designed the experiments, performed the experiments, analyzed the data, prepared figures and/or tables, authored or reviewed drafts of the article, and approved the final draft.
- Cleto Alvarez Aguilar conceived and designed the experiments, performed the experiments, analyzed the data, prepared figures and/or tables, authored or reviewed drafts of the article, and approved the final draft.

### Human Ethics

The following information was supplied relating to ethical approvals (*i.e.*, approving body and any reference numbers):

This study was approved by the Ethics and Research Committees from the Mexican Institute of Social Security (IMSS), which are certified as Institutional Review Board (IRB) in accordance with the Mexican regulation, under protocol number R-2015-785-010.

### Data Availability

The data supporting this research is available in the Supplementary File.

### Supplemental Information

Supplemental information for this article can be found online at http://dx.doi.org/10.7717/peerj.17817#supplemental-information.

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
