# Peer review of "Prevalence and metabolic risk factors of chronic kidney disease among a Mexican adult population: a cross-sectional study in primary healthcare medical units"

_PeerJ, doi:10.7717/peerj.17817_

## Round 0.1 · original submission · Major Revisions

Dear authors

Two reviewers have deeply reviewed your manuscript and raised several critical concerns.

Please read and attend to each one, make the corresponding changes, and redact the rebuttal letter, including the page and line where the changes (if applicable) are highlighted.

**Language Note:** The review process has identified that the English language must be improved. PeerJ can provide language editing services - please contact us at [email protected] for pricing (be sure to provide your manuscript number and title). Alternatively, you should make your own arrangements to improve the language quality and provide details in your response letter. – PeerJ Staff

Reviewer 1 ·

Basic reporting

The submitted manuscript describes a cross-sectional study on adults from a distinct geographical area with the aim of assessing the prevalence and risk factors of chronic kidney disease (according to its KDIGO definition) in this population.
The manuscript is clearly written, in a scientific style, supported by sufficient references from the literature, (both classic and recent). The English language is acceptable, so the manuscript is easily understandable. Also, the manuscript respects the structure of an scientific writing.
As a point which can be improved, I suggest to exclude the old terminology of "microalbuminuria" (for example on page 8 line 102, page 9 line 129 & 167, and anywhere it is repeated along the manuscript) and replaced it with the categories defined by the 2012 KDIGO guidelines (moderately increased albuminuria).

Experimental design

The study hypothesis, even it is not novel, it worth be investigated especially because regional differences could exist, and the Hispanic population was reported to be at a higher risk for both metabolic diseases and chronic kidney disease. The aims of the study are well defined and the research methodology is clearly described, so it would be easy to replicate it. As significant strengths, I emphasize the sample size calculation for statistical power and the repeated analyses (GFR and albuminuria) for a correct diagnosis of chronic kidney disease (as per KDIGO guidelines).
Some clarifications are, however, needed:
- How was "urinary tract infection" diagnosed based on urinalysis strips? Please add mare details on this matter in the Methods section.
- Please add to page 10 line 179: "... persistent at two measurements at three months apart.", as it appears to be the case based on the description of exclusion criteria, a couple of paragraphs earlier.
The statistical methods are appropriate, but the presentation of the results as median (minimum, maximum) is atypical. Please use the median (95% confidence interval) instead. Also, the logistic regression model should be presented in detail (the dependent variable, the independent variables entered in the first step of the model and how were selected, the goodness of fit of the model, how was co-linearity avoided), since only this model can support the associations between metabolic risk factors and the CKD diagnosis.

Validity of the findings

The Discussions and conclusions are supported by the study results. However, improvements of the Discussion section are warranted:
- Please explain the very high prevalence of CKD in the investigated cohort, almost double than the globally reported prevalence. Maybe some selection bias, or local particularities in term of dietary customs or genetic predisposition etc.
- The Discussion section is a little too long, so I suggest to compress the comments and to focus the discussion on the CKD and its risk factors rather than separating the section in distinct paragraphs for each investigated factor.
- Please add a paragraph to acknowledge the study limitations.

Additional comments

In the Results section, some values are duplicated in the text and in the Tables. Please avoid any repetition.

Reviewer 2 ·

Basic reporting

'no comment

Experimental design

'no comment

Validity of the findings

This cross-sectional study surveyed 3,901 adults from primary healthcare units of the Mexican Social Security Institute in Michoacán, reporting a chronic kidney disease (CKD) prevalence of 21.9%. Identified independent risk factors included overweight, obesity, abdominal obesity, hypertension, impaired fasting glucose, type 2 diabetes, dyslipidemia, and hyperuricemia. My comments on the manuscript are as follows:

Major:
1. The study appears to utilize logistic regression to evaluate associations between potential risk factors and CKD without addressing interactions among CKD-associated factors.
2. The study did not specify whether univariate or multivariate logistic regression models were applied. According to Figure 3, sex appears to be adjusted in each logistic regression model. However, if further adjustment for multiple factors was not performed in multivariate logistic regression models, the study cannot conclude that any factor is an "independent" risk factor. To establish independence, the analysis should account for potential confounding variables.
3. As this is a cross-sectional study designed to assess the prevalence of CKD, it cannot establish causality or identify risk factors for the development of CKD.
4. The prevalence of CKD in this study may not be representative of the Mexican adult population, as the participants were recruited from the state of Michoacán. The study should provide more information on how potential participants were selected before invitation and whether the sampling was based on the population structure in Michoacán.
5. The sample size calculation, as stated in the methods section, does not appear to be associated with the primary outcome of the present study. The authors mentioned "The sample size was calculated using the cohort studies formula with a confidence level of 95%, an estimated odds ratio (OR) >1, a specific relative precision of 10%, and a proportion of glucose metabolism alterations of 15%". The sample size calculation should be based on the expected prevalence of CKD and the desired level of precision.

Minor:
1. To accurately estimate glomerular filtration rate (GFR) using the CKD-EPI equation and define CKD according to the KDIGO guideline, standardization of creatinine measurement is critical. The study should provide more details regarding the measurement of serum creatinine. Additionally, it is important to clarify whether the creatinine measurement had calibration traceable to isotope dilution mass spectrometry (IDMS), as this ensures the accuracy and comparability of the results.
2. As the study used the CKD-EPI creatinine equation to calculate estimated GFR (eGFR), the version (2009 or 2021) and the reference for the equation should be provided. This information is necessary for readers to assess the appropriateness of the equation used and to facilitate comparison with other studies.

---

## Round 0.2 · accepted · Accept

I confirm that the authors have addressed the reviewers' concerns and improved the manuscript.

The manuscript is ready for publication.

Reviewer 1 ·

Basic reporting

The suggestion that I previously made on this point (correction of the term "microalbuminuria" was resolved by the authors).
I have nothing more to add.

Experimental design

The authors made the changes that I pointed out in the first round of review. I consider the current version as satisfactory on this respect.

Validity of the findings

The Discussion and Results sections have been improved by the authors, as suggested. No further comments.

Additional comments

No comment